# Does Maternal Mental Health and Maternal Stress Affect Preschoolers’ Behavioral Symptoms?

**DOI:** 10.3390/children8090816

**Published:** 2021-09-16

**Authors:** María Pía Santelices, Matías Irarrázaval, Pamela Jervis, Cristian Brotfeld, Carla Cisterna, Ana María Gallardo

**Affiliations:** 1School of Psychology, Pontificia Universidad Catolica de Chile, Santiago 7820436, Chile; cabrotfe@uc.cl (C.B.); amgallar@uc.cl (A.M.G.); 2School of Medicine, Universidad de Chile, Santiago 8330015, Chile; mirarrazavald@u.uchile.cl (M.I.); ccisternas@fen.uchile.cl (C.C.); 3Universidad de Chile, Institute for Fiscal Studies and Center for Research in Inclusive Education, 8330015 Santiago, Chile; pjervisr@uchile.cl

**Keywords:** maternal depression, parental stress, behavioral problems

## Abstract

(1) Background: Maternal stress and depression are considered risk factors in children’s socioemotional development, also showing high prevalence worldwide. (2) Method: Participants correspond to a longitudinal sample of 6335 mother/child pairs (18–72 months), who were surveyed in 2010 and then in 2012. The hypothesis was tested with SEM analysis, setting the child’s internalized/externalized problems as dependent variable, maternal depression as independent variable, and stress as a partial mediator. (3) Results: Both depression during pregnancy and recent depression has not only a direct effect on the internalizing and externalizing symptomatology of the child, but also an indirect effect through parental stress. Significant direct and indirect relationships were found. (4) Conclusions: Maternal depression and the presence of parental stress can influence children’s behavioral problems, both internalizing and externalizing.

## 1. Introduction

Given the high prevalence of depression in woman, and the fact that this disease also affects three times more women who are raising children [1], we can consider maternal depression as a widespread public health problem that affects the well-being of women and their families [2].

In Chile, there is evidence of high prevalence of female’s depression [3] as well as high rates of parental stress [4].

The association between stress and depression has been previously studied [5,6]. Current studies emphasize a multi-factor conception of parenting stress, involving children and parents’ traits, and sociocultural context including social determinants like poverty [7,8,9]. Parental stress risk increases as a function of the income quintiles of caregivers, with socioemotional development risks being greater in the first quintile: at 6 months there is a 20.4% risk, which increases to 21.2% at 12 months and to 33.7% at 18 months [10].

The relevance of the topic is twofold. It has first a theoretical relevance, which is to identify how parental stress and maternal depressive symptomatology affect socioemotional development and behavioral problems in early childhood. Secondly, it has a practical relevance, which is to validate the importance of caregivers’ wellbeing. Finally, it is also important to generate more local research information, which will help policymakers to design preventive interventions.

From this perspective, we seek to answer the following research question: How do mothers’ depressive symptomatology and parental stress influence children’s socioemotional development and behavioral problems in early childhood?

### 1.1. Depressive Symptomatology in Mothers and Its Impact on Children’s Socioemotional Development

We use the term depressive symptomatology to refer to the behavioral manifestations that configure a mood disorder characterized by a depressed state that manifests itself through a loss of pleasure or interest, changes in appetite or weight, sleepiness, low energy, guilt, difficulties in thinking, concentrating, or making decisions, and thoughts about death, among other features appearing during the last two weeks [11].

There is evidence of the effects of depressive symptomatology on children’s upbringing and interactions with their parents, which includes social models, attachment, monitoring, and discipline [12].

There is also evidence that mothers’ depression is related to changes in family functioning through inadequate role models [13]. Mother’s depression may also be associated with the relationship with the child’s father or life partner [14], which could also affect the child relationship.

According to the attachment theory, when a mother is depressed, the relationship with her child is affected through low sensitivity and availability. Similarly, according to neurobiological theories, in periods of critical growth—such as the first years of life—during which important social, affective, and cognitive information is acquired, maternal failure to provide stimulation will have certain repercussions on the socioemotional area [15]. From the perspective of children’s socioemotional development, maternal depression can be regarded as a risk factor that affects both socioemotional adjustment and social acceptance [13,16] causing poor socioemotional development and behavioral problems such as aggression and a defiant oppositional attitude [12].

A study of 290 mothers seeking professional advice at primary care centers in Chile estimated that 51.9% of the children of depressed mothers had psychiatric symptoms [15]. Among these children, most reported anxious symptoms (62.7%) and other depressive symptoms (25.9%). In addition, 49.8% of the children displayed behavioral and emotional problems. This highlights the importance of prevention in children’s mental health.

Finally, research shows that certain characteristics of children, such as health problems or irritability, could affect a mother’s depression. In other words, a mother’s concerns about a difficult or sick infant may affect her state of mind and worsen both her symptomatology and her child’s difficulties [17]. Children with a difficult temperament are the most likely to develop behavioral problems, although it is worth noting that these issues vary according to children’s age and their interactions with the environment [18,19]. Therefore, if a mother is depressed, she could affect her child’s modulation.

### 1.2. Parental Stress in Mothers and Its Impact on the Child’s Socioemotional Development

The definition of parental stress includes parents’ negative feelings regarding their own parenting skills and also negative feelings towards their offspring [20], causing them to be overwhelmed by the demands in their parenting role [9]. However, it is also important to consider the context, given that parental stress may be worsened by life difficulties such as low socioeconomic status [21] and mothers’ lower educational level [22].

Children of families that cope with continuous distress during children’s first three years of life could be more exposed to develop emotional and behavioral difficulties, but also develop parent-child relationship difficulties [21]. Specifically, mothers with high levels of parental stress tend to have depressive symptoms, anxiety, and an external locus of control, and often display dysfunctional and even abusive educational practices [23,24,25].

Additionally, there is a relationship between children’s development and parental stress. This evidence is significant for socioemotional development if we consider that “problems detected in school have a marked emotional aspect and only by combining the Intellectual Quotient with the Emotional Quotient can good results be achieved” [26,27]. Moreover, Long, Gurka, and Blackman [28] associate parental stress with language acquisition and behavioral problems in young children, while other studies show that children living with very stressed parents are more likely to display internalizing and externalizing problems as well as emotional regulation difficulties [25].

On the other hand, it has been proposed that boys tend to display more externalizing problems than girls [29,30,31], which suggests that mothers with male children might suffer from more parental stress.

### 1.3. Vulnerable Contexts, Children’s Socioemotional Development, and Maternal Mental Health

Socioeconomic status (SES) is related to the quality of the family environment, which is poorer in lower-stratum families. Khawaja, Barazi, and Linos [32] show that when maternal economic difficulties limit mothers’ attention to their children’s adequate nutrition and health, their state of mind and perception of self-efficacy regarding their maternal competences are affected. In this regard, the Organization for Economic Co-operation and Development (OECD) has stated that Chile’s educational problems are linked to its severe social segmentation and that education quality and equity are affected in lower SES populations [33].

According to Rodriguez and Muñoz [34] who analyzed the results of 5005 families who participated in the ELPI (Longitudinal Early Childhood Survey) in Chile, showed that 8.2% of the children were at risk of language and socio-emotional delay. Low maternal educational background and low educational quality at home were important risk factors in children’s development.

As for mental health, The National Health Survey ENS 2009–2011 shows a higher prevalence of depressive symptomatology in women and in people with a lower educational level [35].

Additionally, Ulloa, Cova, and Bustos [36] analyzed a sample of 9996 boys and girls between 3 and 5 years and their caregivers who participated in the ELPI in Chile, and they established a mediation model using Parental Distress as mediator in the relation between SES and internalizing problems and a moderation model using parental distress as moderator in the relationship between low SES and externalizing problems. This result highlights the relevance of including both contextual factors and also caregivers’ mental health in addressing children’s development.

## 2. Methods

### 2.1. Participants and Procedures

Participants were 6335 mother/child pairs drawn from the ELPI. The data used are a subsample taken from the first two rounds of the ELPI (2010 and 2012) [37]. The first round of the survey includes children born between 1 January 2006, and 31 August 2009. In the second round of the survey, the decision was made to extend the sample to include children born between 1 September 2009, and 31 December 2011. Thus, the initial longitudinal sample comprising nearly 15,000 children grew by about 3000 subjects in the second round. The sample is representative at a national, urban, and rural level.

For this paper, we selected homes where the biological mother lives with the child, which represent 98.8% (*n* = 15,754) of the homes surveyed in the first round (2010) and 98.2% of those surveyed in the second round (2012). Then, we linked the first and second round databases, which yielded a sample of 12,611 cases measured at two points (2010 and 2012). Given that the main aim of this study was to understand how parental depression and stress can influence the child’s internalized and externalized symptomatology, we selected homes where child symptomatology was measured with the Child Behavior Check List 1 (CBCL1). This was achieved by selecting those children who were within the age range of the instrument (18 to 72 months) at each of the two points when it was administered. A total of 7181 cases were identified. Finally, 822 cases, corresponding to mothers who decided not to answer the parental stress instrument, were discarded, and 24 cases were left out because these mothers reported having received no formal education whatsoever.

### 2.2. Sociodemographic Data

The mothers comprising this sample are between 17 and 56 years old (M = 31.62, SD = 7.09), while their children are between 39 and 72 months old (M = 57.86, SD = 8.45). Like most other surveys in developing countries, the ELPI provides very detailed information on household assets; thus, it is necessary to reduce the dimensionality of the SES information provided to create a SES score which captures the underlying socioeconomic status of the household. To do this, we used a polychoric principal component analysis, following the approach laid out in Kolenikov and Gustavo [38]. This allowed us to construct a SES score combining continuous, categorical, and discrete variables to estimate the underlying SES factor for each household without violating any assumptions of normality or losing any information associated with a standard principal component analysis. Table 1 presents the variables that were used as controls in the analysis. As shown in Table 1, the standardized SES score is broken into five evenly populated quintiles.

**Maternal depression.** Maternal depression was determined through these three questions: “Were you diagnosed with depression during pregnancy”, with 11.5% (*n* = 730) of affirmative answers; “after pregnancy, were you diagnosed with postpartum depression by a specialist?”, with 12.5% (*n* = 780) of affirmative answers; and “have you recently been diagnosed with depression by a specialist?”, with 12.0% (*n* = 758) of affirmative answers. The last question was made only in the year 2012. Using these data, we created three variables: depression during pregnancy, postpartum depression, and current level of depression.

**Child Externalizing and Internalizing Behavior.** Child behavior was measured by The Child Behavior Checklist 1 (CBCL/1 1⁄2–5). This instrument is administered to the child’s caregivers, who were asked to rate the degree to which they believe each item on the CBCL is true about their child’s behavior within the past 2 months on a scale from 0 (not true) to 2 (often true). The CBCL includes two broadband scales, including in total 99 items. The first, labelled “Internalizing” problems (36 items) consists of four syndrome subscales (Emotionally Reactive, Anxious/Depressed, Somatic Complaints, and Withdrawn). Examples of these items are: “Clings to adults or too dependent”, “Diarrhea or loose bowels (when not sick)”. The second scale, “Externalizing” problems (24 items), consists of two syndrome subscales (Attention Problems and Aggressive Behavior). Examples of these items are: “Can’t concentrate, can’t pay attention for long”, “Hits others”. Standardized T scores are used to estimate a child’s level of impairment relative to the population and cut points have been prescribed for children with scores falling into the “borderline” (93rd percentile) and “clinical” (98th percentile) ranges.

The CBCL/1.5–5 Inventory is an internationally recognized instruments for evaluating maladaptive behaviors that may affect the present and future development of preschool children between 1.5 and 5 years of age (c being supported in numerous studies [39,40], and also validated in Chile by Lecannelier and colleagues [39], and therefore considered useful to study the prevalence of mental health problems in Chilean children.

In this sample, the internal consistency estimated with Cronbach’s alpha for the internalizing problems scale was 0.82 and 0.86 for the 2010 and 2012 measures, respectively. In the case of the scale of externalized problems, the estimated internal consistency was 0.87 and 0.91 for the same years.

**Parental stress**. Parental Stress was assessed using the Edinburgh Parental Stress Index-Short Form (PSI-SF) [7]. This instrument provides a measure of stress levels in a short time of administration, assuming that stress may be due to situational variables, characteristics of the parents, and/or behavioral traits of children that are related to the role of parenthood. It is aimed at the parental domain and consists of 36 items with a Likert-type response scale. Scores are related to three factors/subscales (12 items each): Parental Distress (PD), referred to stress reported by the mother in relation to her personal characteristics associated with the role of motherhood (e.g., “Since having my child I have been unable to try new and different things”); Dysfunctional Parent-Child Interaction (PCDI), which accounts for the stress perceived by the mother in her interaction with her child (e.g., “My child smiles at me much less than I expected”); and a difficult child score (DC), related to whether the mother finds it easy or difficult to control the child according to his/her behavioral traits (e.g., “There are some things my child does that really bother me a lot”). The sum of the three subscales yields an overall score indicating the level of stress that the mother experiences when exercising motherhood [20]. With respect to the instrument’s reliability, the short version in Spanish shows an adjustment above 0.95.

This study uses the PD scale because it has been shown that higher parenting stress is related to caregivers’ depressive emotion [41]. Additionally, PD is shown to be related to negative parenting behaviors such as negative parenting practices and negative emotional responsiveness [42], which are related to children to poor socio emotional development [43]. In this sample the internal consistency estimated with Cronbach’s alpha was 0.87.

### 2.3. Missing Cases

Table 2 presents the amount of missing data separated by variable, which reveals that data loss is especially severe for the parental stress variable. Given that there is great loss of data for some study variables, the sample that has complete data was compared with the original sample, in order to ensure that both samples are similar. In Table 3 both samples are compared, showing similarity between the original sample and the sample that will finally be used to evaluate the hypotheses of the study.

## 3. Results

### Analysis of the Hypothesized Model

To evaluate the presented model, a structural equations model was used, using the lavaan statistical package [44]. The weighted least squares mean and variance adjusted (WLSMV) estimator was used as it is an estimator that allows adjusting ordinal items, as is the case with Likert scales. Estimators designed to adjust ordinal variables are recommended in the case of Likert scales, especially if they have asymmetric distributions or are made up of five or fewer categories [45]. Sobel’s method [46] was used to estimate the standard errors of the indirect effects, in which the betas and standard errors of the direct effects were used. This method has been shown to have less statistical power than resampling methods [47]; however, it was preferred because resampling generates computational demands that could not be covered in the present study.

The model uses two measurements of internalized and externalized problems, one of which was done in 2010 and the other in 2012. The purpose of incorporating two measurements is to control for the baseline state of internalized and externalized problems of the child, so that to determine if the other variables of relevance to the study, specifically parental stress and the various forms of depression, are capable of influencing the child’s symptoms once it has been controlled for what appears to be more stable in the child’s development, and therefore less dependent on context. On the other hand, the sociodemographic variables already presented as control variables were used.

The explanatory model of externalized problems (see Figure 1) had a good fit: X^2^ (5870) = 51811.053, *p <* 0.001, CFI = 0.930, TLI = 0.946, RMSEA = 0.038, SRMR = 0.057. Regarding the hypothesized associations, it is observed that the presence of depression in pregnancy, the presence of a recent depression diagnosis, and the scores of internalized and externalized symptoms measured 2 years earlier, are positively associated with parental stress once sociodemographic variables were controlled for. It is also possible to observe a stability in the scores of externalized problems since the previous score of externalized problems is strongly related to the later externalized problems. Furthermore, it is observed that parental stress, the presence of a diagnosis of depression during pregnancy, and the diagnosis of recent depression are associated with the externalized problems of the child.

Table 4 presents the regression coefficients of the explanatory variables of parental stress including the control variables, which were omitted from Figure 1 to simplify the reading. It is observed that the higher the educational level and the higher the socioeconomic level, the lower the level of parental stress. At the same time, work occupation, with respect to work inactivity, is associated with less parental stress.

Table 5 also presents the associations between externalized problems and covariates. The previous externalized problems are strongly associated with the externalized problems measured two years later, and that the mother’s age is negatively associated with the externalized problems of the child. Similarly, it is observed that the female sex, compared to the male, presents lower levels of externalized problems. Finally, it is not observed that the mother’s socioeconomic level and educational level are directly associated with the externalized problems of the child, being lower in quintile V, compared to the first quintile, and lower for mothers who have completed higher education, with respect to those with basic education.

Regarding the mediation hypothesis, it was observed that internalized and externalized problems, as well as depression in pregnancy and recent depression, are associated with externalized problems through parental stress as shown in Table 6.

The explanatory model of internalized problems (see Figure 2) had a good fit: X^2^ (7256) = 63739.96, *p* < 0.001, CFI = 0.895, TLI = 0.916, RMSEA = 0.038, SRMR = 0.060. Since this model shares the same predictors on parental stress from the externalized problems model, the beta coefficients are practically the same (up to the third decimal) and the difference in R^2^ is only 0.002, (R^2^ = 0.178 in the externalized problem model, and R^2^ = 0.180 in the internalized problems model), so the Table 4 provides the same information regarding the predictors of parental stress.

Regarding the hypothesized associations, it is observed in Table 7 that the mother’s parental stress and the presence of depression during pregnancy are associated with a greater presence of internalized problems, once we had controlled for the sociodemographic variables and for the internalized and externalized problems measured 2 years before. It is also possible to observe a stability in the scores of internalized problems, since the previous score is strongly related to the scores of later internalized problems. Furthermore, it is observed that parental stress, the presence of a diagnosis of depression during pregnancy, and a recent diagnosis of depression are directly associated with the internalized problems of the child.

Regarding the mediation hypothesis, it was observed that internalized and externalized problems, as well as depression in pregnancy and recent depression, are associated with internalized problems through parental stress as shown in Table 8. Thus, although a direct effect of the mother’s recent depression towards the child’s internalized problems was not found, an indirect effect was observed through its association with the mother’s parental stress.

## 4. Discussion

This paper aims to shed light on how maternal depression and the presence of parental stress can influence children’s behavioral problems, both internalizing and externalizing.

It was observed that the presence/absence of maternal depression during pregnancy has not only a direct effect on the internalizing and externalizing symptomatology of the child, but also an indirect effect through parental stress. Numerous studies associate depression with difficulties in parenthood, since it generates a distance in the bond of early attachment [48].

Additionally, parental stress and depression during pregnancy is associated with more externalizing and internalizing problems, even when controlling for sociodemographic variables and measurements of behavioral problems made two years earlier. Children of mothers who report having had depression during pregnancy are more likely to display externalizing problems than children of mothers who did not have depression during pregnancy. Furthermore, recent depression is associated with externalizing and internalizing problems. This highlights the relevance of maternal mental health in children’s emotional development.

However, depression diagnosed during pregnancy was found to directly affect children’s behavior. This is not surprising, given that numerous studies show an association between depression in the last trimester of pregnancy and regulation problems in children [49]. Pregnancy is a vulnerable moment for babies, in which they are exposed to hormonal changes of the mother. During this crucial stage of growth, in which the fetus’ nervous system develops, maternal depression is especially problematic.

Regarding parenthood differences by child gender, it has been shown that mothers are more sensitive to their daughters than to their sons [50], probably because they achieve greater empathy with a child of the same sex. Another aspect is cultural socialization. In Chile, girls learn to be more self-regulated, presenting minor behavioral problems and thus facilitating parenting [51]. This study shows that boys tend to express more externalizing problems than girls, therefore the authors suggest that parental distress is more associated to behavioral problems than children gender.

It is also worth noting that the mother’s SES and educational level were found to have an influence on the child’s internalizing and externalizing problems. This finding can be taken to suggest that these are distal variables whose influence can depend on proximate variables such as upbringing patterns. This is relevant, because it makes it possible to presume that—despite the difficulties and costs of modifying the material and cultural conditions of families—it is possible to adopt strategies aimed at the agents who directly participate in children’s upbringing, at least at the intervention and public policy levels.

However, it is important to point out the limitations of this study. First, all the instruments used by ELPI are self-report which could bias the results, especially regarding the reporting of behavioral problems in children when mothers are diagnosed with depression. This is because depressed mothers could perceive their children as being more problematic. Linked to this, it is important to have in mind how hard it may be for a depressive mother to assess her child’s behavior correctly, because this could also affect her perception of the child and in the future. Second, it would be interesting to see in future studies if the behavioral effects on the child are long-term and if they have an impact on their development, by evaluating the child and not just the mother itself. Then, future studies should try to use observational or multi-informants to evaluate child behavior more objectively. Additionally, it would be interesting to analyze in further studies the link between the SES indicator and lower levels of parental stress in women, as it is seen in more traditional cultures/environments that work for women is considered as a limitation to the time dedicated to a child; hence, there could be families with low incomes where women do not work because of that more traditional idea.

Finally, it is important to highlight that the results of this research article are useful to encourage new public policies that address the relation between caretakers’ mental health and children’s socioemotional development.

## 5. Key Messages

Maternal mental health affects children’s socioemotional development and is related to internalizing and externalizing problems in early infancy.It is fundamental to address maternal mental health in public health service during child medical assistance and in educational system.

## Figures and Tables

**Figure 1 children-08-00816-f001:**
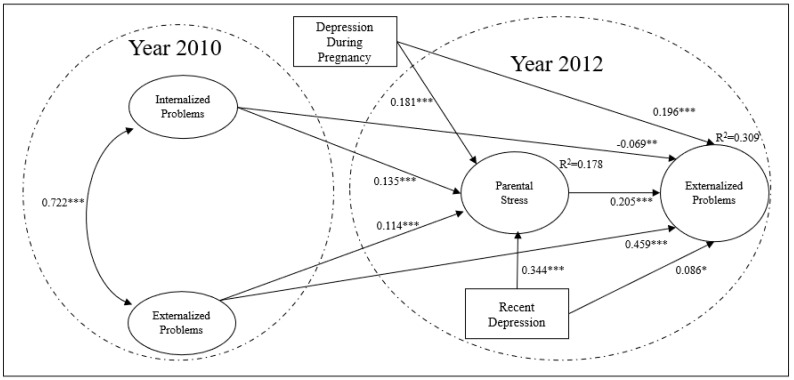
Explanatory model of externalized problems. “***” *p* < 0.001, “**” *p* < 0.01, “*” *p* < 0.05. The standardized βs are presented for the latent variables, and the non-standardized βs for the observed variables. Only the statistically significant associations of the most relevant variables from the theoretical point of view are presented in this study.

**Figure 2 children-08-00816-f002:**
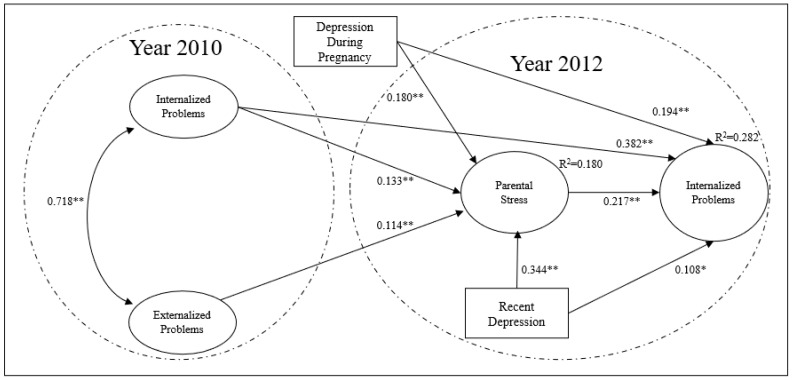
Explanatory model of internalized problems “**” *p* < 0.001, “*” *p* < 0.05. The standardized βs are presented for the latent variables, and the non-standardized βs for the observed variables. Only the statistically significant associations of the most relevant variables from the theoretical point of view are presented in this study.

**Table 1 children-08-00816-t001:** Descriptive statistics.

	N = 6335
Sex of the child	
Boy	3172 (50.1%)
Girl	3163 (49.9%)
Occupational status	
Inactive	2879 (45.4%)
Unemployed	318 (5.0%)
In employment	3138 (49.5%)
Educational level	
Incomplete schooling	1079 (17.1%)
Full schooling	3987 (63.1%)
Incomplete higher education	552 (8.7%)
Full higher education	701 (11.1%)
SES	
Quintile I	1199 (18.9%)
Quintile II	1265 (20.0%)
Quintile III	1265 (20.0%)
Quintile IV	1359 (21.5%)
Quintile V	1247 (19.7%)

Due to its small size, it was determined that it would not be possible to make inferences about this specific group. SES scores were constructed using discrete variables (electrical appliances such as refrigerator, washing machine, DVD, microwave, boiler, video camera, cellphone with a plan, broadband, PC, laptop, paid cable connection) and categorical variables such as type of activity in the labor market, type of flooring materials, and rent payment categories.

**Table 2 children-08-00816-t002:** Number of missing cases per variable.

	Cases with Complete Data	Cases with Missing Data	% of Missing Data
Parental stress	5655	680	10.73%
Internalized problems, 2012	6073	262	4.13%
Externalized problems, 2012	6077	258	4.07%
Postpartum depression	6254	81	1.27%
Internalized problems, 2010	6316	19	0.29%
Externalized problems, 2010	6316	19	0.29%
Recent depression	6317	18	0.27%
Scholarship	6319	16	0.25%

Only the variables for which there are missing data are presented.

**Table 3 children-08-00816-t003:** Comparison between initial sample and sample with complete cases.

	Initial Sample (N = 6335)	Sample with Complete Data (N = 5302)	*p* Value
Sex of the child			0.948
Boy	3172 (50.1%)	2658 (50.1%)	
Girl	3163 (49.9%)	2644 (49.9%)	
Occupational status			0.623
Inactive	2879 (45.4%)	2390 (45.1%)	
Unemployed	318 (5.0%)	258 (4.9%)	
In employment	3138 (49.5%)	2654 (50.1%)	
Educational level			0.061
Incomplete schooling	1079 (17.1%)	824 (15.5%)	
Full schooling	3987 (63.1%)	3375 (63.7%)	
Incomplete higher education	552 (8.7%)	495 (9.3%)	
Full higher education	701 (11.1%)	608 (11.5%)	
Socioeconomic status			0.516
Quintile I	1199 (18.9%)	948 (17.9%)	
Quintile II	1265 (20.0%)	1044 (19.7%)	
Quintile III	1265 (20.0%)	1054 (19.9%)	
Quintile IV	1359 (21.5%)	1170 (22.1%)	
Quintile V	1247 (19.7%)	1086 (20.5%)	
Depression during pregnancy			0.676
No	5474 (87.5%)	4627 (87.3%)	
Yes	780 (12.5%)	675 (12.7%)	
Postpartum depression			0.727
No	5605 (88.5%)	4702 (88.7%)	
Yes	730 (11.5%)	600 (11.3%)	
Recent depression			0.871
No	5559 (88.0%)	4671 (88.1%)	
Yes	758 (12.0%)	631 (11.9%)	

Chi-square test was used to evaluate the differences between the samples.

**Table 4 children-08-00816-t004:** Set of predictors of parental stress.

Parental Stress	se	*p*-Value	β
Mother’s age	0.002	0.476	0.001
Child’s age (months)	0.002	0.497	−0.001
Child’s sex ^1^	0.030	0.074	−0.048
Internalized problems, 2010	0.026	0.000	0.133
Externalized problems, 2010	0.025	0.000	0.114
Depression during pregnancy ^2^	0.048	0.000	0.180
Recent depression ^2^	0.046	0.000	0.344
Postpartum depression ^2^	0.047	0.089	0.073
Unemployed ^3^	0.074	0.262	−0.075
In employment ^3^	0.033	0.000	−0.172
Quintile 2 ^4^	0.048	0.001	−0.145
Quintile 3 ^4^	0.049	0.000	−0.157
Quintile 4 ^4^	0.050	0.000	−0.249
Quintile 5 ^4^	0.055	0.000	−0.505
Secondary education ^5^	0.042	0.000	−0.327
Incomplete higher education ^5^	0.067	0.000	−0.564
Full higher education ^5^	0.067	0.000	−0.594
R^2^	0.178		

“^1^” Boys are used as a reference point. “^2^“ Lack of depression is used as a reference point. “^3^” Inactive mothers are used as a reference point. “^4^” The first quintile is used as a reference point. “^5^” People who only attended primary school are used as a reference point.

**Table 5 children-08-00816-t005:** Set of predictors of externalized problems.

Externalized Problems 2012	se	*p*-Value	β
Parental stress	0.016	0.000	0.205
Internalized problems	0.028	0.003	−0.069
Externalized problems	0.028	0.000	0.459
Mother’s age	0.002	0.000	−0.017
Child’s sex ^1^	0.032	0.000	−0.177
Child’s age (months)	0.002	0.000	−0.010
Recent depression ^2^	0.051	0.043	0.086
Depression during pregnancy ^2^	0.055	0.000	0.196
Postpartum depression ^2^	0.051	0.470	0.031
Quintile 2 ^3^	0.054	0.704	−0.017
Quintile 3 ^3^	0.054	0.365	−0.041
Quintile 4 ^3^	0.055	0.270	−0.050
Quintile 5 ^3^	0.060	0.011	−0.127
Secondary education ^4^	0.048	0.009	−0.104
Incomplete higher education ^4^	0.073	0.026	−0.135
Full higher education ^4^	0.071	0.001	−0.192
R^2^	0.309		

“^1^” Boys are used as a reference point. “^2^“ A lack of depression is used as a reference point. “^3^” The first quintile is used as a reference point. “^4^“ People who only attended primary school are used as a reference point.

**Table 6 children-08-00816-t006:** Indirect effects through parental stress on externalized problems.

	se	*p*-Value	β
Internalized problems	0.006	<0.001	0.028
Externalized problems	0.006	<0.001	0.023
Depression during pregnancy	0.012	<0.001	0.037
Recent depression	0.011	<0.001	0.070

**Table 7 children-08-00816-t007:** Set of predictors of internalized problems.

Internalized Problems 2012	se	*p*-Value	β
Parental stress	0.017	0.000	0.217
Internalized problems	0.033	0.000	0.382
Externalized problems	0.028	0.430	−0.019
Mother’s age	0.003	0.000	−0.009
Child’s sex ^1^	0.034	0.711	0.011
Child’s age (months)	0.002	0.676	−0.001
Recent depression ^2^	0.051	0.013	0.108
Depression during pregnancy ^2^	0.056	0.000	0.194
Postpartum depression ^2^	0.051	0.221	0.053
Quintile 2 ^3^	0.055	0.594	−0.025
Quintile 3 ^3^	0.056	0.013	−0.118
Quintile 4 ^3^	0.056	0.007	−0.130
Quintile 5 ^3^	0.063	0.000	−0.233
Secondary education ^4^	0.018	0.000	−0.186
Incomplete higher education ^4^	0.028	0.000	−0.310
Full higher education ^4^	0.027	0.000	−0.356
R^2^	0.282		

“^1^” Boys are used as a reference point. “^2^” A lack of depression is used as a reference point. “^3^” The first quintile is used as a reference point. “^4^” People who only attended primary school are used as a reference point.

**Table 8 children-08-00816-t008:** Indirect effects through parental stress on internalized problems.

	se	*p*-Value	Std. lv
Internalized problems	0.006	<0.001	0.029
Externalized problems	0.006	<0.001	0.025
Depression during pregnancy	0.004	<0.001	0.039
Recent depression	0.005	<0.001	0.075

## Data Availability

Data base from year 2010 available at http://observatorio.ministeriodesarrollosocial.gob.cl/elpi-primera-ronda (accessed on 20 July 2021) and data base from year 2012 available at http://observatorio.ministeriodesarrollosocial.gob.cl/elpi-segunda-ronda (accessed on 20 July 2021).

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
