# Peer review of "Does Maternal Mental Health and Maternal Stress Affect Preschoolers’ Behavioral Symptoms?"

_children, 2021, doi:10.3390/children8090816_

Round 1

Reviewer 1 Report

The article deals with an important problem of the determinants of externalizing and internalizing disorders in children. The determinants were searched for in selected SES factors and the mother's psychological health, namely depression and parental stress.

The aim was to check the research - as one might expect - of a longitudinal nature (due to the measurement of internalizing and internalizing disorders in children approximately two years apart. However, questions arise whether these studies are longitudinal? in children of different ages (can the symptoms of disorders be expected to be the same?) In a child at the age of e.g. 6 or 12 months and in a child aged 30 or 36 months, or in older children? disorders, but they are not the same.

The authors assumed that the occurrence of symptoms of depression in a woman / mother during pregnancy and depression in the perinatal period and in the "last" period (the term was not specified) has a negative effect on the child's development and behavior. However, the repetition of depression may indicate certain 'constitutional' factors and may not necessarily be related to the parental role.

Only mothers were examined and the family is the system! It is worth considering narrowing the category / concept of parental stress to maternal / maternal stress or maternal parental stress

It is worth mentioning this in the article.

The article was given a social dimension as it emphasized the importance of the socio-economic situation of a woman for the effectiveness of her functioning as a mother and the child's behavior. This sensitizes readers primarily to poor economic conditions and their consequences for the upbringing of younger generations, and thus for the health of the society.

Some specific issues:

How is parental stress understood?

Understanding well-being? What welfare is it about? For a sense of psychological well-being? Emotional? Well-being as measured by economic indicators? It is not clear from the rest of the article that these are objective, economic and social indicators.

Narrowing down the current depressive state to two weeks - it seems risky due to the undertaken problem - inferring about the child's development.

Apart from the mother, the family often includes the child's father, siblings and other family members (e.g. grandmother) - they can replace the mat in caring for the child, they may also be depressed, etc.

Depression in the mother - may be associated with the child's father, life partner, so it is worth paying attention to the relationship between the child's mother and his father / mother's partner - for example in the introduction to the topic.

The functioning of the family and the performance of family roles has a cultural context - this is what the authors of the article point out. Perhaps it is worth writing briefly about the specificity of the culture in Chile? About the environments of the respondents, the genealogy of the respondents, indigenous peoples? immigrant population?

The authors mention modulation. As far as modulation is concerned, the importance of hereditary conditions cannot be ruled out.

Is the parent able to accurately define his parenting skills, especially in the case of people with a lower level of education? This may also be related to the level of cognitive processes, intellectual abilities, mental resilience, resilience, etc.

Could it be assumed that the mother's depression affects her in the manner assigned to her? Even if depression occurs during pregnancy and / or postnatal depression is diagnosed after the birth of the baby, are the effects on the child's behavior long-term? - after all, behavioral disorders in the child have been studied in the last 2 months. Also, questions were asked about the symptoms of depression in mothers in the recent period of time - this gives the basis for (possible) inference about - perhaps a “temporary nature” of the child's behavioral disturbances caused by the mother's current behavior. Besides: is a depressed mother able to correctly assess the child's behavior?

This may be highlighted in the discussion of results, in the paragraph on study limitations, or as issues requiring clarification in further studies.

The question arises as to the link between the SES indicator - such as work - with lower levels of parental stress in women: in traditional environments, work may be perceived as limiting the time devoted to a child. Has a woman's professional activity been linked to her education? With the family income level? There are probably families where incomes are not low - and the woman does not work because of that.

No info with R value - in the table from Figure 1 it follows that R = .178

How to understand in the article that mothers are more sensitive to daughters than to sons? Cultural diversity?

children sex? is it about gender?

It would be beneficial for the reception of the content of the article - to refine it a bit.

Author Response

Reviewer 1

Responses

The aim was to check the research - as one might expect - of a longitudinal nature (due to the measurement of internalizing and internalizing disorders in children approximately two years apart. However, questions arise whether these studies are longitudinal? in children of different ages (can the symptoms of disorders be expected to be the same?) In a child at the age of e.g. 6 or 12 months and in a child aged 30 or 36 months, or in older children? disorders, but they are not the same.

The participants correspond to a longitudinal sample who were surveyed in 2010 and then in 2012.

In the study, is aimed to analyze if internal and external symptoms depend on maternal depression or not, but not to analyze if child symptoms are expected to be the same.

The authors assumed that the occurrence of symptoms of depression in a woman / mother during pregnancy and depression in the perinatal period and in the "last" period (the term was not specified) has a negative effect on the child's development and behavior. However, the repetition of depression may indicate certain 'constitutional' factors and may not necessarily be related to the parental role.

Could be. We will mention this in the manuscript.

Only mothers were examined and the family is the system! It is worth considering narrowing the category / concept of parental stress to maternal / maternal stress or maternal parental stress

It is worth mentioning this in the article.

Addressed.

The title has been changed to maternal stress, instead of parental stress.

How is parental stress understood?

The definition of parental stress includes parents’ negative feelings regarding their own parenting skills and also negative feelings towards their offspring (Díaz-Herrero, Brito de la Nuez, López, Pérez-López & Martínez-Fuentes, 2010), causing them to be overwhelmed by the demands in their parenting role (Webster-Stratton, 1990). However, it is also important to consider the context, given that parental stress may be worsened by life difficulties such as low socioeconomic status (Oliva, Montero, & Gutiérrez, 2006) and mothers’ lower educational level (Olhaberry & Farkas, 2012).

Understanding well-being? What welfare is it about? For a sense of psychological well-being? Emotional? Well-being as measured by economic indicators? It is not clear from the rest of the article that these are objective, economic and social indicators.

Well-being understood as psychological well-being and linked to depressive symptomatology.

Depression in the mother - may be associated with the child's father, life partner, so it is worth paying attention to the relationship between the child's mother and his father / mother's partner - for example in the introduction to the topic.

Addressed.

Mother’s depression may also be associated with the relationship with the child’s father or life partner (Lindhorst & Oxford, 2008), which could also affect the child relationship.

The functioning of the family and the performance of family roles has a cultural context - this is what the authors of the article point out. Perhaps it is worth writing briefly about the specificity of the culture in Chile? About the environments of the respondents, the genealogy of the respondents, indigenous peoples? immigrant population?

This is addressed in the introduction and discussion sections.

Participants were obtained from the ELPI survey. The sample was representative at a national, urban, and rural level. Results form families who participated in the ELPI (Longitudinal Early Childhood Survey) in Chile, showed that 8,2% of the children were in risk of language and socio-emotional delay. Low maternal educational background and low educational quality at home were important risk factors in children development.

Is the parent able to accurately define his parenting skills, especially in the case of people with a lower level of education? This may also be related to the level of cognitive processes, intellectual abilities, mental resilience, resilience, etc.

In this study we do not measure parenting skills, so we do not have that information.

Could it be assumed that the mother's depression affects her in the manner assigned to her? Even if depression occurs during pregnancy and / or postnatal depression is diagnosed after the birth of the baby, are the effects on the child's behavior long-term? - after all, behavioral disorders in the child have been studied in the last 2 months. Also, questions were asked about the symptoms of depression in mothers in the recent period of time - this gives the basis for (possible) inference about - perhaps a “temporary nature” of the child's behavioral disturbances caused by the mother's current behavior. Besides: is a depressed mother able to correctly assess the child's behavior?

This may be highlighted in the discussion of results, in the paragraph on study limitations, or as issues requiring clarification in further studies.

Addressed.

The question arises as to the link between the SES indicator - such as work - with lower levels of parental stress in women: in traditional environments, work may be perceived as limiting the time devoted to a child. Has a woman's professional activity been linked to her education? With the family income level? There are probably families where incomes are not low - and the woman does not work because of that.

Interesting point, also added to the discussion part.

Also, it is interesting to analyze in further studies the link between the SES indicator and lower levels of parental stress in women, this seen as the possibility in more traditional cultures/environments that work in women can be seen as a limitation to the time dedi-cated to a child, hence there could be families with low incomes were women do not work because of that more traditional idea.

No info with R value - in the table from Figure 1 it follows that R = .178

Addressed.

How to understand in the article that mothers are more sensitive to daughters than to sons? Cultural diversity?

In the article we express that “mothers are more sensitive to their daughters than to their sons (Hallers-Haalboom et al., 2014), probably because they achieve greater empathy with a child of the same sex. Another aspect is cultural socialization. In Chile, girls learn to be more self-regulated, presenting minor behavioural problems and thus facilitating parenting (Olhaberry, 2012)”. So it could be cultural diversity or it could be not. But it was worth to mention it.

children sex? is it about gender?

Addressed.

Changed to children gender.

Reviewer 2 Report

This manuscript analyzes as maternal mental health and parental stress is related to child behavioral symptoms. Sample was 6,335 Chilean mothers between 17 and 56 years old (M = 31.62, SD = 7.09) with one child between 39 and 72 months old (M = 57.86, SD = 8.45), tested in 2010 and 2012. Parental stress seem that was measured with the parental distress scale (authors not specify any item example).  Maternal depression was measured with depression during pregnancy (‘Were you diagnosed with depression during pregnancy’), postpartum depression (‘after pregnancy, were you diagnosed with postpartum depression by a specialist?’) and current depression (‘have you recently been diagnosed with depression by a specialist?’). Preschoolers’ behavioral symptoms was measured with internalized (authors not specify any item example) and externalized problems (authors not specify any item example). Main findings indicated that maternal depression during pregnancy and recent depression, on one hand, have a direct influence on child internalizing and externalizing symptomatology, but on the other hand, also have an indirect influence through parental stress. In conclusion, maternal depression and the presence of parental stress can influence children’s behavioral problems, with both internalizing and externalizing symptoms.

Overall the work is very good. The most neglected part corresponds to the instruments. It is necessary to provide examples of items and to calculate the internal consistency at any given moment (e.g., internalizing and externalizing symptoms in 2010 and 2012).

Author Response

Reviewer 2

Response

Overall the work is very good. The most neglected part corresponds to the instruments. It is necessary to provide examples of items and to calculate the internal consistency at any given moment (e.g., internalizing and externalizing symptoms in 2010 and 2012).

Addressed.